# Ethical inclusion: Risks and benefits of research from the perspective of perinatal people with opioid use disorders who have experienced incarceration

Julia Reddy[1]*, Kristel Black[2], Keia Bazemore[3], Kiva Jordan[3], Jamie B. Jackson[3], Andrea K. Knittel[3]

1 Department of Maternal Child Health, Gillings School of Global Public Health, University of North Carolina at Chapel Hill, Chapel Hill, NC, United States of America, 2 University of North Carolina at Chapel Hill School of Medicine, Chapel Hill, NC, United States of America, 3 Department of Obstetrics and Gynecology, University of North Carolina at Chapel Hill School of Medicine, Chapel Hill, NC, United States of America

* jreddy@email.unc.edu

## Abstract

### Background

Research ethics guidelines and emphasis on representation in research guide the inclusion of marginalized groups, including people with perinatal opioid use disorders (OUD) and people experiencing incarceration in the United States. However, insights from participants regarding the risks and benefits of participation are not adequately considered. The aim of this study was to examine the risks and benefits of research participation from the perspective of pregnant/postpartum people with OUD who have experienced incarceration.

### Design

We recruited people who had experience with perinatal incarceration and were either currently pregnant or postpartum, and at least 18 years old. All participants met the clinical criteria for OUD. Our study did not have exclusion criteria based on gender, race, or ethnicity.

### Setting

Participants were either currently incarcerated at the North Carolina Correctional Institute for Women in Raleigh, North Carolina, United States or had previously experienced perinatal incarceration and were recruited from a perinatal substance use disorder treatment program located in North Carolina.

### Participants

Between 9/2021-4/2022, we completed 12 interviews with pregnant/postpartum people with OUD, approximately half who were currently incarcerated and half with a recent history of perinatal incarceration.

**Data Availability Statement:** The sensitive nature of the data precludes the sharing of the dataset via a public repository. Investigators interested in

conducting analyses with this qualitative data should contact the North Carolina Department of Adult Corrections (administrative-analysis@ncdps.gov) and the senior author in order to initiate a process of obtaining appropriate approvals from the University of North Carolina at Chapel Hill Institutional Review Board and the North Carolina Department of Adult Corrections Prison Research Review Committee.

**Funding:** AK was funded by Grant # K12 HD103085, University of North Carolina Women's Reproductive Health Research Career Development Program, PI Neal-Perry. The funders had no role in study design, data collection and analysis, decision to publish, or preparation of the manuscript.

**Competing interests:** The authors have declared that no competing interests exist.

## Intervention/measurement

Interviews were conducted via Webex phone or video. The interviews followed a scripted interview guide and lasted one hour on average. Interview transcripts were analyzed using the Rigorous and Accelerated Data Reduction technique to produce an overarching thematic framework.

## Findings

Our analysis identified benefits, including the personal advantage of self-expression, helping others and contributing to change, and financial incentives. Risks included stigma and breach of confidentiality, misunderstanding of the distinction between research and advocacy, and limited ability to share their whole experience.

## Conclusions

Participant-identified benefits of research mirrored those from other marginalized populations, though participant-identified risks were novel and nuanced. Recruitment and consent should move beyond normative research ethics committees protocol language to consider the perspectives of participants.

## Introduction

In 1978, the United States' National Commission for the Protection of Human Subjects of Biomedical and Behavioral Research outlined ethical guidelines for conducting research with human subjects (the Belmont Report). The Belmont Report has guided determinations of ethical research practices through institutional review boards and research ethics committees with its basic moral principles of respect for persons, autonomy, beneficence, and justice [1, 2]. While the Belmont Report arose from a recognition of decades of coercive experimentation that preceded the convening, some have suggested that research has now proliferated and become commodified to a point where a reconsideration of guiding ethical principles is needed [3]. Specifically, an analysis of the Belmont Report through a feminist critical lens points out the role of oppression and marginalization in both an increased vulnerability to exploitation in research and also underrepresentation or exclusion from research studies [4]. Systematic underrepresentation in research, as that experienced routinely by women and pregnant individuals denies these populations any potential benefits of study inclusion and perpetuates gaps in data that prohibit generalization of outcomes and improvement of interventions [4, 5].

Establishing methodologies that maximize ethical inclusion of vulnerable populations is essential [6]. Informed consent is the primary process through which potential participants in research learn about the potential harms as well as the potential benefits they may accrue through research participation. One consequence of the intense regulatory focus on clinical trials is near consensus on the acceptable risks and benefits of receiving experimental therapies, with a concurrent shift away from regulatory scrutiny of social and behavioral science research practices [2]. Institutionalized ethics procedures run the risk of becoming perfunctory, without considering, from the perspectives of those who have been historically underrepresented or excluded, the unique benefits and risks of which potential research participants should be made aware in order to give informed consent [7].

Particularly in settings of confinement, including prisons and to some degree substance use treatment centers, ethical vigilance is needed. It has been noted that confined individuals are highly controlled and constrained, physically and in access to activities, time, privacy, and autonomy [8]. In the United States (U.S.), statute restricts research of people confined in prisons to studies designed to investigate phenomena or experiences affecting confined individuals as a distinct population or that has a reasonable probability of improving their wellbeing [9]. However, a tension exists between protectionism, or restricting the study of populations that are deemed vulnerable to coercion, and fair inclusion in ethical research [10]. In this paper, we exemplify the merit of soliciting participant input to understand the benefits and harms of qualitative research participation, as well as specific barriers to research participation experienced by populations with intersecting marginalized identities.

Recent inquiries into the ethical considerations of research, particularly with marginalized populations in the U.S., have shown discrepancies between how participants view their participation in research studies and how they are typically presented in ethics review boards or study protocols [11]. For instance, compensation for research participation is seen as an incentive for voluntary participation in most study protocols, while it may be seen as paid work by participants [12]. Other research found that a highly marginalized population of HIV-positive Black women gained personal insight from research participation and also valued their potential to help others through participation in the study [13]. Close to 100 individuals who had survived rape, when asked about their experience participating in in-depth interviews overwhelmingly talked about positive, cathartic, and productive feelings arising from the interview process, which employed feminist interview strategies including methods to reduce the hierarchical disparity between interviewer and respondent [14].

Understanding the perspective of research participants is vital to improve methodologies as well as to effectively recruit and engage important populations in studies. Strategies for the inclusion and protection of populations who are considered vulnerable to coercion or exploitation by research have been outlined by previous scholars. For instance, collaboration and dialog between researchers, healthcare providers who serve vulnerable populations, and members of the target populations may increase transparency and improve voluntary study engagement [15]. Participation on research teams by individuals affected by the systems or circumstances that are the research foci improves study recruitment and engagement [16]. In addition to soliciting participant feedback to inform future study designs, there is merit in incorporating participant input iteratively into ongoing study methodologies and engagement strategies [17].

There is a need to strategize the inclusion of individuals with intersecting marginalized identities in research studies in ways that are participatory and ethical. We contribute to this important literature by relaying the perceived risks and benefits of research participation among a U.S. population of individuals who have experienced the intersecting vulnerabilities of pregnancy, incarceration, and substance use disorder.

## Materials and methods

### Eligibility

We recruited people who were at least 18 years old, met the clinical criteria for opioid use disorder (OUD), and had experienced perinatal incarceration in the past year. Although most of our participants identified as women/female, our study did not have exclusion criteria based on gender, race, or ethnicity. Participants were provided with paper consent forms to read while the study coordinator reviewed them via Webex. A staff member not involved in the participants' medical care provided the consent form to the participant. Due to prison rules, this staff member was required to be present in the room during consent. Our study protocol

included a confidentiality statement signed by the person present at consent and this limit of confidentiality was included in the consent form. Afterwards, the PI collected the signed consent forms, and they were later signed by the study coordinator.

## Setting

Approximately half of participants were currently incarcerated at the North Carolina Correctional Institute for Women (NCCIW) in Raleigh, North Carolina. Within the NCCIW, there is a prenatal clinic that provides care to all pregnant or postpartum individuals in the prison. There are 15–30 pregnant or postpartum people at the prison at any given time and about half meet criteria for OUD. The remaining participants had previously experienced perinatal incarceration and were recruited from a perinatal substance use disorder treatment program located in North Carolina. This study project was approved by the Institutional Review Board at the University of North Carolina at Chapel Hill School of Medicine (# 20–3559).

## Recruitment and enrollment

We recruited participants for the study at both NCCIW and the treatment program between September 2021 and April 2022. Recruitment at NCCIW consisted of providing clinic nurses with postcard-sized flyers to share study information. Potential participants who were interested in learning more about the study were directed to a private space in the prenatal clinic which they could ask study staff questions about the study; this interaction occurred either just before or just after a prenatal visit such that clinic and facility staff would not be able to distinguish between receiving usual clinical services or learning more about the study. At the treatment program, we presented the study very briefly at several group sessions. Information at these sessions was similar to the flyers. Potential participants were directed to contact the study team directly to learn more about the study. We did not enumerate the number of people outreached to who declined to participate.

## Consent

Following screening and enrollment, participants were offered time prior to consenting to participate in the study. The informed consent process was conducted via Webex phone or video by the study coordinator (JJ). For participants who were incarcerated at the time of the interview, an additional non-clinician member of the research team observed their Webex use inside the facility in a room where they could not be observed or overhead by carceral facility staff. Participants who were not incarcerated were instructed to call or connect via video from a private location. The study coordinator reviewed the consent form, which described details of the study as well as potential risks and benefits of participation, and answered questions (S1 File). Participants were informed that the goal of the research was to learn about the challenges facing women who use drugs and are pregnant or just had a baby while they are in prison, and specifically that the study aimed to understand what makes women decide to be in a research study or not to be in a research study, and what might make it easier for them to stay in a study when they go home. All potential participants were informed that their participation would not affect prenatal care during or after incarceration or any aspect of parole or incarceration. For participants at NCCIW, the additional non-clinician staff member witnessed the participant's signature on the consent form. Participants not at NCCIW provided verbal consent as all study procedures were conducted via video and phone. Of note, in order to avoid a real or perceived conflict of interest, the senior author was not involved in the consent process as she is also a prenatal care provider at the prison.

## Interviews and safety monitoring

Interviews were conducted via Webex phone or video by the female-identified study coordinator (JJ) with assistance from two notetakers; notetaker identities were anonymous and their videos were turned off during the interviews. The interviews followed a scripted guide and lasted one hour on average.

To optimize participant safety and wellbeing, we took several specific steps. Our research team had personal and clinical experiences with substance use disorder and incarceration and were equipped to perform their study roles respectfully and empathetically. Our primary interviewer was a clinical research coordinator, doula, and childbirth educator, and is adequately trained to conduct interviews and provide emotional support and referrals.

The interview guide (S2 File) was designed from a trauma-informed perspective, rooted in a Reproductive Justice theoretical framework [18, 19], and included important domains from the literature and the research questions. The main objective of the interview questions that inform this analysis was to inform robust recruitment and community retention processes for people with OUD from a prison prenatal clinic. As such, the interview questions asked specifically how a participant heard about the study; why they chose to participate in the interview; what things were considered during that decision-making process; what, if anything would have made them want to participate more or less; and whether they would be willing to participate in a follow-up interview–if that were a possibility. While they were explicitly told that the current study did not include follow up interviews, they were asked how study staff might best contact them after transition back into the community, in the hypothetical case of a follow up interview. Currently incarcerated participants did not receive any financial incentive for participating–per prison policy–and non-incarcerated participants were compensated with $50 for their interview.

## Data analysis

The interview transcriptions were taken in the form of contemporaneous notes, with an emphasis on verbatim documentation of participant response. The first four interviews were pilot interviews, after which a review of the interview guide was completed. As no major changes were made, the pilot interviews were included in our total sample. The research team analyzed the transcripts using the Rigorous and Accelerated Data Reduction (RADaR) technique [20]. Using a generalized inductive approach, the codebook was developed *a priori* with topical codes from the interview guide and then was expanded with interpretive codes identified during open coding of initial interviews. Each interview was coded independently by 2–3 coders and disagreements were resolved through discussion. We completed a second round of coding after we identified two additional interpretive codes that had not been previously captured. Two members of the research team (AK and JR) drafted an overarching thematic framework which was revised through discussion with the entire research team until inductive thematic saturation was reached [21]. Themes are displayed in Table 1.

## Results

In total, we conducted 12 interviews. We grouped the emergent themes regarding research participation from these interviews into two groups: (a) benefits of participation in qualitative research interviews, and (b) risks of participation. Exemplative quotes are included below with assigned numbers distinguishing each research participant in a way that minimizes risk of deductive disclosure.

**Table 1. Thematic framework developed through a generalized inductive method.**

| Themes | Sub-themes | Codes |
|---|---|---|
| Benefits | *Therapeutic sharing* | Get my word out, Research interest, Emotional aspects |
| | *Sharing for others* | Helping others, Making change |
| | *Participation as autonomy* | Autonomy |
| | *Financial benefits* | Research Interest, Financial incentive |
| Risks | *Confidentiality* | Confidentiality |
| | *Misunderstanding of the scope and pace of the research* | Helping others, Making change |
| | *Lack of autonomy to share the full story* | Longitudinal perspective, Study communication: Phone; Social media messaging; Text messaging; Email; Postal mail; Personal contacts; Barriers |

## Benefits

We asked participants to share their decision-making process when deciding to participate. There was a good deal of consistency in the participants' expressions of value and motivation related to why they decided to participate in the research study. Thematic responses related to the personal advantage of self-expression, helping others and contributing to change, and the incentives in terms of autonomy and compensation.

**Therapeutic sharing.** Participants consistently spoke of the value of being able to talk about their experiences in incarcerated settings and navigating OUD during the perinatal period with a perceived neutral party.

One participant explained:

*It's also good to be able to tell my story and have someone be interested and understand without judgement.* (Participant #1)

Another participant concurred:

*Cause I just want to get some stuff off the chest. Let it out in a safe place.* (#2)

When asked why she decided to participate in the study, one person said:

*My drug use and being pregnant, that's something I wanted to talk about. It's something I can't get too honest about, if someone asked me about it. I wanted to talk about it, because addicted to drugs and being early in my pregnancy: I wanted to share that experience and tell you how I felt about it, and how everything turned out okay.* (#3)

**Sharing for others.** The most common benefit to study participation expressed by interviewees was the opportunity to help others and potentially contribute to positive change, particularly in the treatment of perinatal women and those with OUD in incarcerated settings.

Multiple participants talked about wanting to share how bad prison was for them:

*I wanted to be a part of it to share my experience. Some women have a different experience, and it might be a good experience but my experience of being in prison and pregnant has not been good. It has not been "peaches and cream" by far and I wouldn't wish it on my worst enemy. Feels like people should know what pregnant women in prison go through. Particularly those that are coming off of drugs.* (#4)

A similar sentiment, expressing a desire to help spare others from what they have experienced, was expressed by a different participant:

*I want to kind of give a voice like a different opinion than the same stuff that we hear sometimes like 'that lady is getting treatment'. [Prison is] not all that it's meant to be. I like to talk and see if I can help people. I don't want somebody to be in my shoes, I wouldn't wish this on anybody, so if I can talk or help somebody, [I want to].* (#5)

The motivation to help others seemed to directly influence participants' decision to enroll in the research study:

*Another girl from here, we were cell mates in prison and pregnant together and she told me about it and that's why I called. And I called immediately as soon as she told me 'cause I thought our stories could help. I was all in as soon as I heard about it.* (#6)

**Participation as autonomy.** Participants in highly structured residential treatment programming and those experiencing incarceration experience restrictions on autonomy. Multiple participants spoke of their schedules, insinuating a lack of control over how they spent their time. There was a sense that, alongside a desire to participate in the research, the study was an option among a limited number of choices for how to spend time.

*But I like helping. Because if you can't ask us your questions than you can't do your research. I'm a helper. And it got me out of group for an hour. It's nice to change gears and do something different.* (#7)

**Financial benefits.** Participants talked about benefits to participation in the form of financial incentive (which was only provided to non-incarcerated participants). Importantly, research participation by those who were not currently incarcerated was seen by some participants as a reliable way to make money:

*Well since I've been pregnant, I've done a few research studies, that was what I was doing to get extra money.* (#8)

Another person described the benefit of the monetary compensation as filling an essential need, in a situation with little access to financial support:

*The second aspect [in deciding to participate] is the financial incentive, because people like me–I don't have the resources to provide for my kids, so I have to rely on the treatment facility to graciously provide groceries for me once a month, and that is something I am so grateful for.* (#9)

## Risks

When asked about their decision-making around research participation, themes emerged from participants' responses that highlighted several nuanced risks of participation.

**Confidentiality.** The research focus on participants who experienced addiction raised fears of breach of confidentiality for some, though not all, participants. Despite communication of the confidentiality and data protections characterizing the study protocol, some participants mentioned hesitation related to whether their names would be used and where the information would end up.

*When I first heard about it, I was skeptical if it was going to be a lot of people. I was scared of how it was going to go, if I had to go somewhere or there would be a lot of people.* (#2)

Two other participants concurred, citing relational or perceptional risks in the case of a breach of confidentiality:

*The confidentiality aspect would have had weight in my decision because I don't know where it's going to be used, because if direct quotes were used or names were dropped, I'd have to go after you guys for royalties–no, because my family would see it, and they're not all supportive.* (#9)

*I'm just trusting that you won't take this information and use it for anything other than what you told us you were using it for, because I am getting out one day.* (#10)

Others did not see potential disclosure as a barrier:

*Honestly, I don't care, I have nothing to hide. Y'all could of took my real name, and made a whole exhibit out of it, I wouldn't have given a damn. But people are really about their privacy and stuff.* (#10)

**Misunderstanding the scope and pace of the research.** A key risk was related to how some participants described their understanding of the purpose of the study in the context of their motivation to participate. Although the consent process included a description of the study that emphasized learning about challenges facing participants and understanding their decisions to participate or not in research generally, some of their responses implied confusion between advocacy and research in terms of the scope and pace of change that could come from participation. Multiple interviewees talked about participating out of a desire to make change, which was described above as a benefit of participation. However, some participants spoke of their participation in the study as an opportunity to advocate or make change more directly and in a shorter timeframe. One participant said,

*I really want to advocate for there to be a shift in how we handle the rehabilitation aspect. And I hope that one day I will have the means and ability to help people who are in this situation, so this is an opportunity for me to help so that my experience isn't in vain.* (#9)

In fact, women repeatedly described the research project as important to making change and implied an interpretation of our research goal as changing the prison system.

*. . .The system is set up and designed for our failure and the way we treat and look at addicts is outdated–there are people out there in the world that feel the same and are trying to make change. . .So [I participated because I am] really just wanting a change for that and wanting a change for the next person. I hope it never happens but if it were my children, I would want it to be different.* (#9)

While many participants spoke of a real and urgent hope that their participation would result in immediate, tangible system change, others acknowledged the iterative and slow nature of research.

*I think it will help, if not me, then someone eventually. Y'all are doing this research for a reason. Not just being nosy, not just sitting there at a computer all day for the hell of it. I like to*

*think there's other people in the world that want to make a change and take baby steps for that. So I'd like–if there's anything I can do that may be helpful. . . I have a whole life in me. That's a living breathing life. She's gonna be here for a long time, long after I'm gone. And this prison's gonna probably still be here. and it needs some changes*! (#10)

Some participants showed a high level of understanding of the connections between research aims and institutional systems change:

*Because I feel like the more people that do it and answer as honestly as possible their experience then eventually things might get better or change and resources will be there for pregnant women and women incarcerated, after they get out. . . I'd rather do [the study] than not do it. I like to think it's gonna make a different some day and not just for me, but for–when I get older things might be different and we can make the world a better place.* (#10)

Another interviewee commented similarly:

*Yeah, because if I tell what happened to me, it may not make a big difference but if a couple of people do this study, then maybe the word will get out there and people might notice and say something needs to be done. Hopefully the study will change something at some point.* (#11)

**Lack of autonomy to share the full story.** Although this study methodology consisted of a single interview, participants were asked about hypothetical follow-up interviews and facilitators and barriers of continued participation. Interestingly, multiple interviewees talked about the potential value of longitudinal research participation, in terms of being able to capture the outcomes or changes that take place in their lives over time. Participants talked about adding richness or information to the research be being able to report how things turned out, or what life was like for them post-release or after residential treatment.

*I probably would [do a follow-up interview] just so that way you guys could see the difference between being in prison and coming home.* (#12)

There was a strong sense of the value of longitudinal research, expressed by several participants:

*I definitely would [do a follow-up interview] just because it would–you're talking to me now, you're getting some answers from me now, let's say 3, 4, 5 months down the road, I'm out– that would help the study, see how my answers may have changed, or how I've changed. It would help the study to get an update on how I'm doing.* (#10)

*I can think of a lot of ways that that would help, once I'm out–once a person is out–you know, 'how's it going*?', *you're different this time.* (#10)

Inherent in these comments, and particularly for participants who were incarcerated at the time of the interview, was an implied unknowing or unpredictability about their future. They repeatedly expressed a lack of surety about what would happen to them, where they might end up, how they might be doing post-release, and how they could be contacted, once back in a community. Some of this uncertainty related to the outcomes of their pregnancy:

*Yes [I would do a follow-up interview] to let you know that what I been doing it didn't affect the baby, or it did, or how was it. Just to reconnect and update you that everything is still good. I would want to, to update.* (#2)

There was also uncertainty about the result of their charges or legal case, with implications for their lives and their ability to participate in longitudinal research:

*Cause I'm interstate: I violated in [State 1] and in [State 2]. Honestly, I'm not going to be on probation once I leave the prison–at least in [State 1]. So that's one thing. It's more of a nervous than an exciting thing. Knowing I've handled everything here, and can handle things [in State 2], but it's more stressful than exciting.* (#3)

*I'm leaving on nine-month parole. I do have a pending charge in another county, but I might get it dismissed because the warrant's messed up and the dates that they have I was incarcerated, so I hope that works out in my favor.* (#4)

Uncertainty was expressed regarding where they may be living. Poignantly, one participant exemplified how the uncertain outcome of a treatment program application will determine the city where she lives:

*I'm trying to venture off to [town], to the [treatment program]. I've put in applications for that. I think I got accepted, so that's where I intend to locate. After that, if [town] treats me nice, I'll stay in [town].* (#4)

Areas of uncertainty included most spheres of their post-release lives, including: what support network might be available back in the community:

*I was living with my boyfriend before and he's in jail and I don't know if he'll be in or out of jail by the time I'll get out, and my baby will be with my mom, so I'm going to live with my mom, where my baby's at.* (#12)

Whether they would be able to sustain their sobriety:

*I'm more worried about trying to keep my mind busy so I don't slip and come back. So, trying to stay busy and get back to the things I used to do before all this happened.* (#5)

Where they would live:

*I've done other rehabs and once you finish the rehab it's just boom and you're out. But now I have a baby so I can't just leave here and go sleep on someone's couch.* (#11)

*Public housing–they don't do it for families anymore. You can still apply but once you have felonies on your record, you won't get approved.* (#3)

And how they would make money:

*Temp agencies have a no-discrimination thing, but the hours would mean I can't spend time with my child. . .I can't begin to think about what shift I would want because I don't know what my baby's schedule would be. I have to figure that out because I know I have to work to pay for diapers, pay for childcare, pay for daycare and everything. . .So that's some challenges but I can make it work, you know. I mean, I have to.* (#3)

Almost all participants, when asked about how a researcher might contact them for a future interview, referred to not having reliable access to a phone or a phone number that didn't

frequently change. This inconstancy seemed to reflect and contribute to an overall uncertainty about their future stability and circumstances.

## Discussion

This analysis identified perceived risks and benefits to qualitative study participation in a sample of incarcerated and previously incarcerated perinatal individuals with OUD, which confirm and extend previous findings.

Consistent with prior literature, our research confirms a three-level beneficial experience of participation as personally cathartic, purposeful in providing a perceived opportunity to help others or improve a negative situation, and meaningful to assist in knowledge generation [22]. Similar to existing analyses, our study demonstrated benefits that were both individual and relational: participants talked about emotional release as well as the desire to connect, advocate for a cause, and help others in similar situations [23, 24]. These interview responses offered insight into study benefits not typically included in informed consent processes and reflect participants' reasoning for participation during the interview. Research into the experience of those being interviewed differed in the assessment of risk and benefit from normative ethics review protocol language, underscoring the importance of incorporating the perspectives of research participants during the study protocol [25].

Risks identified in this study included fear of a breach of confidentiality, and participants shared complicated decision-making about how much this risk weighed in their decisions. Additionally, a mixed level of understanding around the boundary between research and advocacy exemplified opportunities to improve the consent process and descriptions of research goals during recruitment. Finally, the repeated reference to a lack of certainty about the future highlights an important potential restriction to equitable participation that must be considered when recruiting marginalized populations in research studies–participants may both prefer longitudinal research designs that allow them to share a more full picture of their lives and the effects of incarceration and also be limited in their participation in such research due to the duration and conditions of incarceration.

The concept of autonomy is defined in the Belmont Report as an individual's right to self-determination around research participation, with full knowledge of the potential risks and benefits of the study [26]. The process of informed consent relies on voluntariness, disclosure, understanding, and capacity [27]. In addition to these four principles, we propose the consideration of availability to participate. Our findings highlight the ways that freedom to participate in research is compromised in the lives of marginalized populations, and challenge research committees to encourage equitable ways to reduce logistical barriers to participation for those unduly burdened.

Intersecting vulnerabilities carry unpredictability and a lack of autonomy to control one's surroundings [28]. In our sample, participants described unpredictability regarding pregnancy, incarceration, stigma, financial and residential security, and sobriety. Future uncertainty, and particularly the additive effects of intersecting uncertainties, have direct implications for research inclusion [29]. Universally, participants expressed lack of autonomy that impacted their ability to be contacted, either because of unstable housing or lack of a reliable phone, computer, or contact. If qualitative studies aim to equitably include members of marginalized populations, these uncertainties must be accounted for in study recruitment and engagement protocols in both simple and complex ways. For example, prohibitions on the use of social media for study communication without prior written or verbal consent via another communication medium may directly limit participation for those without stable addresses or phone numbers. More abstractly, the regulatory and logistical challenges that investigators

face when including participants who are incarcerated or are under supervision/surveillance in the community may deter full recruitment and retention. The inability to confer payment to incarcerated participants to compensate their participation is a barrier to equity and an example of regulations that impede full participation. In this way, the concept of protectionism, which refers to the important limitations around participation by vulnerable populations in order to prevent exploitation, unequally excludes groups with limited autonomy from the benefits of research participation [30]. Our findings affirm this risk and align with a recommended reconsideration of the definition of vulnerability in research ethics protocols in order to reduce restrictions on certain populations' participation [10, 31].

## Limitations

These data are not without limitations. Qualitative studies are, by design, intended to generate highly localized insights, novel hypotheses, and deep understanding of phenomena. They are not generalizable. Thus, our findings should be interpreted in the context of the North Carolina institutions in which they were conducted–a perinatal treatment program and a state prison–even as they raise questions and considerations for a broader context. Conducting research in a prison setting also presents unique challenges. We were unable to provide financial incentive for research participation for incarcerated participants, meaning the mention of financial incentives as a benefit applied only to half of our sample. Although this made it more likely that we would hear from only the most motivated participants, and potentially those with strong positive or negative experiences or opinions, the degree of data saturation and inductive thematic saturation we noted are reassuring [21]. In addition, the prison setting has the high potential for social desirability bias in responses as the interviews were required to be conducted with an additional member of the research team observing in person. To mitigate this, the consent process was very clear that the research team was separate from facility staff and that research participation would not affect the incarceration or medical care. However, some distinctions between the responses of participants living in the treatment setting as opposed to the confinement setting may have resulted from these differences in interview protocol. We were unable to review the transcripts and findings with participants, as we were not able to contact them after the single interview. Finally, we acknowledge that participants were at varying places in their recovery and that creating and sharing recovery-oriented narratives of treatment and reentry is an important part of this process [32–35]. Our data were limited in that we gathered insights from participants at only one point in time during incarceration and during recovery.

## Conclusion

Potential participants should have the liberty and agency to decide about participation in research studies. Classification of pregnant and incarcerated individuals as vulnerable research subjects in need of procedural protections, must be careful not to limit participation of highly marginalized populations [36, 37]. Even when participants do provide consent, circumstances limiting their autonomy may circumscribe their research participation. They may not be able to opt *in* to research participation to the extent that they desire. The differentiation that has been previously established between procedural ethics and ethics in practice [7], particularly in settings of confinement [8], can help organize the participant-derived benefits found in this analysis that may exist outside of formal procedural requirements. While research ethics protocols should be examined according to their effectiveness at protecting vulnerable populations from harm and also fairly permitting those same populations to participate in research, when desired, ethical principles guiding research with vulnerable populations must go further.

Participatory inventories of benefits and risks, as well as a stringent adherence to ethics in practice will enable the responsible inclusion of marginalized populations in high quality qualitative research.

## Supporting information

**S1 File. Consent to participate.**
(PDF)

**S2 File. MATernity recruitment and retention interview guide.**
(PDF)

## Acknowledgments

The authors would like to acknowledge Timelie Horne, Joia Freeman, Kim Andringa, Senga Carroll, Dr. Elton Amos, and the prenatal clinic nursing staff at NCCIW for their invaluable contributions, without which this research would have been impossible.

## Author Contributions

**Conceptualization:** Julia Reddy, Andrea K. Knittel.

**Data curation:** Julia Reddy, Keia Bazemore, Kiva Jordan, Jamie B. Jackson.

**Formal analysis:** Julia Reddy, Kristel Black, Keia Bazemore, Kiva Jordan, Jamie B. Jackson, Andrea K. Knittel.

**Funding acquisition:** Andrea K. Knittel.

**Investigation:** Julia Reddy.

**Project administration:** Jamie B. Jackson, Andrea K. Knittel.

**Supervision:** Andrea K. Knittel.

**Writing – original draft:** Julia Reddy, Kristel Black, Andrea K. Knittel.

**Writing – review & editing:** Julia Reddy, Keia Bazemore, Kiva Jordan, Jamie B. Jackson.

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
