## [Decision Letter · Decision Letter 0]

29 Jun 2023

PONE-D-23-02896New considerations for inclusion in research: Recruitment of pregnant and postpartum people who have experienced incarceration with substance use disordersPLOS ONE

Dear Dr. Reddy,

Thank you for submitting your manuscript to PLOS ONE. After careful consideration, we feel that it has merit but does not fully meet PLOS ONE’s publication criteria as it currently stands. Therefore, we invite you to submit a revised version of the manuscript that addresses the points raised during the review process.

We look forward to receiving your revised manuscript.

Kind regards,

Alan Farrier, Ph.D

Academic Editor

PLOS ONE

Journal Requirements:

This work was primarily supported by the UNC WRHR Career Development Program (K12 HD103085, PI Neal-Perry).

However, funding information should not appear in the Acknowledgments section or other areas of your manuscript. We will only publish funding information present in the Funding Statement section of the online submission form. 

AK was funded by Grant # K12 HD103085, University of North Carolina Women's Reproductive Health Research Career Development Program, PI Neal-Perry. The funders had no role in study design, data collection and analysis, decision to publish, or preparation of the manuscript. 

Reviewers' comments:

Reviewer's Responses to Questions

**Comments to the Author**

1. Is the manuscript technically sound, and do the data support the conclusions?

Reviewer #1: Partly

Reviewer #2: Yes

2. Has the statistical analysis been performed appropriately and rigorously? 

Reviewer #1: N/A

Reviewer #2: N/A

3. Have the authors made all data underlying the findings in their manuscript fully available?

Reviewer #1: Yes

Reviewer #2: No

4. Is the manuscript presented in an intelligible fashion and written in standard English?

Reviewer #1: Yes

Reviewer #2: Yes

5. Review Comments to the Author

Reviewer #1: Thank you for the opportunity to review this manuscript, which presents qualitative data from 12 interviews with pregnant/postpartum people with opioid use disorder. The paper has many strengths, but could be improved in the following ways:

-Please include additional information in the Introduction about the unique ethical considerations of conducting research with incarcerated populations and those in substance use treatment programs. The authors begin to discuss the systematic underrepresentation in research (p. 3), but the background on research in prisons, in particular, is warranted given the focus of the paper.

-Pg. 5, Line 122 - define "recent past"

-In the methods (p. 5), the authors mention that there were several differences between the pregnant/postpartum people who were incarcerated and those in treatment (e.g., compensation, having a prison staff member present). The authors should revisit this in the discussion. What are the potential limitations that follow from having these different (even if necessary) approaches and what are the implications of these limitations for their findings? Some commentary on the inherent ethical issues here seems important, too (e.g., the inequity inherent in being able to pay some participants in this study, but not others despite their equal participation). The authors may find a place for this on page 15 after line 373 or around lines 377-380. Alternatively, this could fit well in an expanded discussion of the issues raised on page 16 (lines 383-385).

-More information is needed on the interview - What were some of the questions that were asked? How was the interview guide developed? Was the interview guided by prior work or a theoretical approach? What were the overall goals of the interview? Were the interview questions that were analyzed for this particular study part of a larger study. If so, please describe. To me, this is currently a major missing piece to this study and it's hard to interpret the results without more detail on the interview itself. For example, the authors note that many participants discussed the "value of longitudinal research" (p. 12) - were they asked specifically about this? or about their willingness to be interviewed in the future? It's just not clear what the results mean without knowing what participants were asked.

-Pg. 6, Line 158 - I think there is a typo in this sentence which makes it unclear; please revise.

-Please check the journal's guidelines for formatting direct quotes of different lengths; in many places I felt there was not enough interpretation/connection between quotes.

-I found some of the quotes hard to follow or fit within one of the themes. For example, uncertainty about getting into a treatment program (p. 13) - where does this fit within the themes?

-Were there any different themes by setting (prison vs. treatment)? It seems like one of the themes (financial benefit) isn't relevant for half of the sample. This should be acknowledged again the limitation of the sample.

-Please clarify/expand what is meant by "Finally, the repeated ...in research studies." (pg. 15, Lines 364-366).

-Please expand on the discussion of "protectionism" and provide relevant citations (pg. 17, line 408).

Reviewer #2: Data is witheld in accordance with IRB approvals and this is acceptable.

General

I have to read this ‘from’ somewhere, and in my case it is the UK (as a feminist qualitative researcher and ethicist with an international hat).

This is a really good paper. I realised quite how good it is, as after the first read through it followed me around for a few days (in the way a song will …), and I think there is something significant here. The core empirical findings and the participants voices are compelling.

I am sure it will chime with many of us doing qualitative feminist research, but also in relation to research ethics. I hope the external eye is useful regarding polishing the clarity of the presentation, to maximise its usefulness.

You will see from the notes below, there are some things missing from the general presentation which would help readers to understand – a few times I had the feeling I had skipped a section where something had been explained already, and went back to check … but I don’t think it was there.

I have some concerns that the framing of ethics and governance issues is not clear enough between US-specific issues, general or wider issues, and thus the related generalisability of the implications for inclusion in research etc.

I always give detailed reviews, so the word count below doesn’t reflect quality. Hopefully it will be clear what I think is needed, all of which is easily addressed, I think.

I would strongly recommend publication with some minor revisions.

Title / keywords

These comments are for the authors to take from as they wish:

I think the authors are underplaying the content slightly in the title and keywords.

The lack of a keyword around the inclusion/perceptions of risk and benefit issues seems to be a gap – if I was looking for research like this (and I do) how would I find this article?

The title is a problem for me – is there too much emphasis on the (researchers’) process of recruitment here? It doesn’t really represent the paper, as ironically one of the things I am asking for is more information about the recruitment process. I think ‘New considerations for inclusion in research’ is rather generic, and the paper isn’t about substance use in general.

Can the authors find a way to say more clearly that it is a study into e.g., something closer to indicating - What the views of people with perinatal opioid use disorders (OUD) who have experienced incarceration can tell us about the benefits and risks of participation in qual research interviews to inform ethical inclusion ...

Things I like

I don’t need to say that it is well-written, but it is.

I like the subtle contribution to the non-identification of participants of ‘approximately half’ in each group.

I also like the seemingly effortless use of non-binary gendered language, which shows how easy it is do this.

I like the quality of the participant data, which tells us a lot about the research team.

And I like that the same code is identified as both a risk and a benefit (Helping Others. Making Change)

I very much like the exploration of what we might call the ‘advocacy/change/impact misconception’ (akin to the ‘therapeutic misconception’ in clinical research.) This is really important, particularly as it is something researchers’ can affect (take responsibility for) in the way we present studies.

Nature of the study

These points need some clarification:

It took me a while to work out that this paper was reporting the whole study – and that it was not an adjunct to a study ‘about’ something else, that had led to gathering research participants’ views on that experience. This is great, and quite innovative I think – we certainly need more robust research (as this is) about this area. But this needs to be a bit clearer (including in the abstract and title) – this is important not just for clarity but because it would highlight that this is a more unusual paper, especially in this criminal justice setting.

There are various places where I thought, ‘But what did they tell the participants they were doing?; What questions did they ask in the interviews?’. Some of this is the inevitable ‘wanting to know more’ when you like something, but it is also about the presentation of the content so readers can orient ourselves to this study and its results. (See below).

US setting of the study

This needs some attention.

This point is not related to generalisability of the results of the qualitative work, rather to the implications being drawn from them for wider ethics and governance frameworks.

I think the authors have two options – either to present this study more clearly as a specifically local/national example, as research in the US via an IRB, with various limitations and implications for improving the situation in the US, which could then be discussed in an international context, or 2. to present the study as situated in the US, but having implications globally – I would prefer the latter, and assume that was your intention, but it currently sits too far in the middle.

The most prominent issue for me is the specificity/limitations of the paper regarding ethics governance in the US. This needs to be addressed for accessibility and usefulness for an international audience, but is straightforward to do.

For example the first sentence of the Introduction – does this whole sentence refer to the situation in the US, or is it only the final clause about incarcerated individuals in the US? I would like to see this setting/starting point more clearly framed from the very beginning.

The Introduction could use a little tweaking to clarify which parts/claims are about the US specifically, and where/at what point it broadens to the wider / international situation. It invokes a lot of pertinent international literature, although I note not much that is very recent and the debate is moving fast (choice of action on that left to the authors). It would be helpful to clarify the specific/generic nature of the argument here, and at what point this shifts from one to the other. Simple signposting would do the trick.

There is a strong dependence on the Belmont Report in framing the paper, which does not have the same function beyond the US (although it is a standard authoritative reference of course). This is more of a comment, which the authors can consider as they wish.

I do want to ask if there are any guidelines in the US regarding research in this setting – statutory/professional/disciplinary etc? Is the Belmont Report the only point of reference for the guidelines mentioned (the rest seems to be academic literature, or possibly guidelines nested in that academic literature)?

International comparisons

This point is about the framing and presentation, which is the main thing that needs some development.

The current recommendations/comments regarding governance have unacknowledged limitations, as there are different practices and expectations in other countries. Also different terminology regarding e.g., Research Ethics Committees as opposed to IRBs. Simply using a combination term (RECs/IRBs) would open this up.

Some of the specific ethics/governance claims about IRB requirements would not apply in other countries, e.g., (381-83). I am not aware of any research situation internationally with a blanket ban on social media recruitment, in fact quite the opposite is true, especially since 2020. So this needs contextualising (was this perhaps study-specific?).

And contra the situation described in the paper, in UK ethics review (and in many other countries and regarding international research codes), researchers are expected to consider the benefits and risks that potential research participants from all groups should be made aware of in order to give informed consent (90-93). This has led to the standard inclusion of ‘Patient and Public Involvement’, community consultation etc (or equivalent)s in research design, which is now a recommendation/requirement of many funders across the world. It is nowhere near perfect or adequate, but this situation doesn’t chime with the statements in the paper that there is no expectation on researchers to do this. Can the authors make it clear what they mean here?

And would any references to the international literature (codes/guidelines) help? If you feel not, then don’t do it.

Does the situation reported relate solely to the individual IRB approach for this study? Or is this more general in USA? This is of real interest to a wider readership, so even a couple of sentences would help to situate this.

What was the informed consent process, and what were participants told?

Informed consent processes are discussed several times, both generically, and specifically for this study. But I would like to be clearer about what material this contained and what the participants’ journeys were.

The incarcerated participants received a flyer, but what did it say? How did they opt in following that? There is a subsequent process, but we don’t know what it is. In a context of incarceration discussing the autonomy of participants, this is crucial to understand. This connects to the data about participants’ decision-making processes, which would be stronger if we knew the context of what that process included for them.

What does the consent form discussed in this study contain? In UK (again as an example but this is common practice elsewhere too) the standard process is that a consent form is a separate document which cross-refers to the Participant Information Sheet (usually constructed against a standard template), which includes all relevant details. So if someone only saw the consent form, they would not have much information. I think some simple more explanatory information here would be helpful for a wide international readership.

The treatment programme participants had a different route into the study. Do we need to know anything more about this differentiation as it seems important? What did the presentations entail? Might it have impacted their understanding of the research? It sounds as though it was more interactive and discursive.

Could any of this recruitment material go in as supplementary material (IRB approval allowing but there should be no reason why this couldn’t be shared)? That would make it easy to just add small points where relevant and cross-refer to the supplementary material.

Results

It would be good to know what the questions for the interview were – e.g., were the participants asked about other experiences of research (as suggested by some of the data)? A couple of sentences would be enough, with or without the schedule as supplementary material.

It would be good to know what reasons participants were given to take part in the study/what the agenda was – and to know how that intersected with their decision-making and the data that emerged. For example on p11 (264-5) there is a reference to the research goal, but we don’t know what that was. Or how it was communicated to potential participants.

343-346 – This is just a comment from practice: I am curious about why there would not be an option for the participants to contact the researcher regarding future studies. It is standard practice in most settings for participants to be given the lead researcher’s (at least) contact details before they even opt-in, and although in practice people rarely use these, they are generally urged to keep the information for their records. So referring to this would be a straightforward way to keep lines of communication open (although the only time I tried it, no-one contacted us even though they said they would …). It may of course not be possible/feasible for incarcerated people to receive/keep this information, but those in e.g. a treatment programme might appreciate it? It would not resolve all the issues of course, but my point is that it is an existing mechanism which could help. No action is required here.

Discussion

Addressing a few points here would make it clearer – some are connected to those raised above.

In my experience, especially in social-science/health type studies, much participant information refers to the benefit of sharing, helping others, improving services etc. Often in ethics review we have to roll this back a bit, to not over-promise to participants the realistic impact of their contributions. Again, is this context-specific? If there are such limitations where the authors are working this is of huge interest to others.

I’m not sure what ‘ascertaining subjective perspectives of research participants during the study protocol’ means (359-60). Is something missing here?

Knowing what participants were told (as above) would help consideration of their understanding of the boundary between research/advocacy as it would provide some context when we get to this central point.

I’m confused about why there is a suggestion to extend ‘capacity’ to situational factors. Capacity is generally understood as literal clinical type capacity to understand, and there is quite rightly much discussion about e.g., not excluding people with dementia, or those with learning disabilities from research on this basis. I doubt it is intended, but currently this point sounds like it plays into the hands of paternal exclusionists, or ‘protectionism’ (385), which the authors say they want to resist, and I am not sure what purpose it serves here. Given the international debate about challenging concepts of vulnerability in research mentioned (388-90 which again seems maybe light on the standard/recent literature, but again is the authors’ decision to act or not) this seems out of step – do you need this here at all? Or if you do want to include it, can you clarify what you mean?

As a tiny point (361) It would be helpful just to clarify if there was actually a risk of breach of confidentiality (which I doubt) or was it a perceived (highly justifiable) concern for participants?

Julie Cook

6. PLOS authors have the option to publish the peer review history of their article (what does this mean?). If published, this will include your full peer review and any attached files.

Reviewer #1: No

Reviewer #2: **Yes: **Dr Julie Cook

---

## [Author Response · Author response to Decision Letter 0]

30 Aug 2023

Journal Requirements

Comment Response Edited Selection

1. Please ensure that your manuscript meets PLOS ONE's style requirements, including those for file naming. Will do – thank you. 

2. Please provide additional details regarding participant consent. In the ethics statement in the Methods and online submission information, please ensure that you have specified what type you obtained (for instance, written or verbal, and if verbal, how it was documented and witnessed). If your study included minors, state whether you obtained consent from parents or guardians. If the need for consent was waived by the ethics committee, please include this information. We have added a section to the methods labeled “Consent” with additional details. Following screening and enrollment, participants were offered time prior to consenting to participate in the study. The informed consent process was conducted via Webex phone or video by the study coordinator (JJ). For participants who were incarcerated at the time of the interview, an additional non-clinician member of the research team observed their Webex use inside the facility in a room where they could not be observed or overhead by carceral facility staff. Participants who were not incarcerated were instructed to call or connect via video from a private location. The study coordinator reviewed the details of the study and the consent form and answered questions. All potential participants were informed that their participation would not affect prenatal care during or after incarceration or any aspect of parole or incarceration. For participants at NCCIW, the additional non-clinician staff member witnessed the participant’s signature on the consent form. Participants not at NCCIW provided verbal consent as all study procedures were conducted via video and phone. Of note, in order to avoid a real or perceived conflict of interest, the senior author was not involved in the consent process as she is also a prenatal care provider at the prison.

3. Funding information should not appear in the Acknowledgments section or other areas of your manuscript. We will only publish funding information present in the Funding Statement section of the online submission form. 

Please remove any funding-related text from the manuscript and let us know how you would like to update your Funding Statement. 

Please include your amended statements within your cover letter; we will change the online submission form on your behalf. We have removed the funding statement from the Acknowledgements. There are no updates to the Funding Statement as this funding was already disclosed there. This work was primarily supported by the UNC WRHR Career Development Program (K12 HD103085, PI Neal-Perry).

Reviewer 1

1. Please include additional information in the Introduction about the unique ethical considerations of conducting research with incarcerated populations and those in substance use treatment programs. The authors begin to discuss the systematic underrepresentation in research (p. 3), but the background on research in prisons, in particular, is warranted given the focus of the paper. Thank you for this important comment. We have added strong supporting literature to both the Introduction and Conclusion. Introduction: Particularly in settings of confinement, including prisons and to some degree substance use treatment centers, ethical vigilance is needed. It has been noted that confined individuals are highly controlled and constrained, physically and in access to activities, time, privacy, and autonomy (Gomes & Duarte, 2020). In the US, statute restricts research of people confined in prisons to studies designed to investigate phenomena or experiences affecting confined individuals as a distinct population or that has a reasonable probability of improving their wellbeing (Lapid, Ouellette, Drake, & Clarke, 2023). However, a tension exists between protectionism, or restricting the study of populations that are deemed vulnerable to coercion, and fair inclusion in ethical research (Friesen et al., 2023).

Conclusion: The differentiation that has been previously established between procedural ethics and ethics in practice, particularly in settings of confinement (Gomes & Duarte, 2020), can help organize the participant-derived benefits found in this analysis that may exist outside of formal procedural requirements. While research ethics protocols should be examined according to their effectiveness at protecting vulnerable populations from harm and also fairly permitting those same populations to participate in research, when desired, ethical principles guiding research with vulnerable populations must go further. Participatory inventories of benefits and risks, as well as a stringent adherence to ethics in practice will enable the responsible inclusion of marginalized populations in high quality qualitative research.

2. Pg. 5, Line 122 - define "recent past" Thank you – we have clarified. MATERIALS AND METHODS

Eligibility and Consent

We recruited people who had experienced perinatal incarceration currently or in the past year, were at least 18 years old, and met the clinical criteria for opioid use disorder (OUD).

3. In the methods (p. 5), the authors mention that there were several differences between the pregnant/postpartum people who were incarcerated and those in treatment (e.g., compensation, having a prison staff member present). The authors should revisit this in the discussion. What are the potential limitations that follow from having these different (even if necessary) approaches and what are the implications of these limitations for their findings? Some commentary on the inherent ethical issues here seems important, too (e.g., the inequity inherent in being able to pay some participants in this study, but not others despite their equal participation). The authors may find a place for this on page 15 after line 373 or around lines 377-380. Alternatively, this could fit well in an expanded discussion of the issues raised on page 16 (lines 383-385). Thank you! Yes, this is important. First, we added explicit mention of the potential for effect of the different protocol approach in our Limitations section.

Second, thank you so much for offering examples of places in the text to mention inherent inequity presented by the carceral setting. We have added a sentence as recommended. DISCUSSION

Limitations

In addition, the prison setting has the high potential for social desirability bias in responses as the interviews were required to be conducted with an additional member of the research team observing in person. To mitigate this, the consent process was very clear that the research team was separate from facility staff and that research participation would not affect the incarceration or medical care. However, some distinctions between the responses of participants living in the treatment setting as opposed to the confinement setting may have resulted from these differences in interview protocol. 

DISCUSSION

The inability to confer payment to incarcerated participants to compensate their participation is a barrier to equity and an example of regulations that impede full participation.

4. More information is needed on the interview - What were some of the questions that were asked? How was the interview guide developed? Was the interview guided by prior work or a theoretical approach? What were the overall goals of the interview? Were the interview questions that were analyzed for this particular study part of a larger study. If so, please describe. To me, this is currently a major missing piece to this study and it's hard to interpret the results without more detail on the interview itself. For example, the authors note that many participants discussed the "value of longitudinal research" (p. 12) - were they asked specifically about this? or about their willingness to be interviewed in the future? It's just not clear what the results mean without knowing what participants were asked. We have added additional information about the interview guide in addition to the theoretical approach (Reproductive Justice) that informed the development of the interview guide. 

 MATERIALS AND METHODS

Interviews and Safety Monitoring

The interview guide was designed from a trauma-informed perspective, rooted in a Reproductive Justice theoretical framework (14, 15), and included important domains from the literature and the research questions. The main objective of the interview questions was to inform robust recruitment and community retention processes for people with OUD from a prison prenatal clinic. As such, the interview questions asked specifically how a participant heard about the study; why they chose to participate in the interview; what things were considered during that decision-making process; what, if anything would have made them want to participate more or less; and whether they would be willing to participate in a follow-up interview – if that were a possibility. While they were explicitly told that the current study did not include follow up interviews, they were asked how study staff might best contact them after transition back into the community, in the hypothetical case of a follow up interview.

5. Pg. 6, Line 158 - I think there is a typo in this sentence which makes it unclear; please revise. Ah, thank you. It is not a typo, but it is unclear! Rephrased: MATERIALS AND METHODS

Data Analysis

The interview transcriptions were taken in the form of contemporaneous notes, with an emphasis on verbatim documentation of participant response.

6. Please check the journal's guidelines for formatting direct quotes of different lengths; in many places I felt there was not enough interpretation/connection between quotes. Thank you! We aimed to use quotes to comment without redundant annotation, but we have added language throughout the RESULTS section that may assist in setting up each quotation. 

7. I found some of the quotes hard to follow or fit within one of the themes. For example, uncertainty about getting into a treatment program (p. 13) - where does this fit within the themes? Thank you – this looks like a place where additional contextualizing language is needed. We have added this. RESULTS

Risks

Lack of autonomy to share the full story

Uncertainty was expressed regarding where they may be living. Poignantly, one participant exemplified how the uncertain outcome of a treatment program application will determine the city where she lives:

8. Were there any different themes by setting (prison vs. treatment)? It seems like one of the themes (financial benefit) isn't relevant for half of the sample. This should be acknowledged again the limitation of the sample. Thank you – yes. We have amended our Limitations section to explicitly mention this. DISCUSSION 

Limitations

We were unable to provide financial incentive for research participation for incarcerated participants, meaning the mention of financial incentives as a benefit applied only to half of our sample.

9. Please clarify/expand what is meant by "Finally, the repeated ...in research studies." (pg. 15, Lines 364-366). Ah – We have reworded for clarity. We also added a distinction in the presentation of that theme to respond to your above point of distinct experiences dependent on setting. RESULTS

Risks

Lack of autonomy to share the full story

Inherent in these comments, and particularly for participants who were incarcerated at the time of the interview, was an implied unknowing or unpredictability about their future… There was also uncertainty about the result of their charges or legal case, with implications for their lives, their ability to participate in longitudinal research, and the ability of researchers to understand their lives:

DISCUSSION

Finally, the repeated reference to a lack of certainty about the future highlights an important potential restriction to equitable participation that must be considered when recruiting marginalized populations in research studies – participants may both prefer longitudinal research designs that allow them to share a more full picture of their lives and the effects of incarceration and also be limited in their participation in such research due to the duration and conditions of incarceration.. 

10. Please expand on the discussion of "protectionism" and provide relevant citations (pg. 17, line 408). Yes- that point needed to be more clearly stated and cited. While we do introduce the concept of protectionism in the introduction, we should still here make a point of returning to it more clearly. We have made the following change in the conclusion. CONCLUSION

Potential participants should have the autonomy to decide about participation in research studies. Protectionism, while well intended, may limit participation of highly marginalized populations (Baldwin, Sobolewska, & Capper, 2020). Even when participants do provide consent, circumstances limiting their autonomy may circumscribe their research participation. They may not be able to opt in to research participation to the extent that they desire. The differentiation that has been previously established between procedural ethics and ethics in practice, particularly in settings of confinement (Gomes & Duarte, 2020), can help organize the participant-derived benefits found in this analysis that may exist outside of formal procedural requirements. While research ethics protocols should be examined according to their effectiveness at protecting vulnerable populations from harm and also fairly permitting those same populations to participate in research, when desired, ethical principles guiding research with vulnerable populations must go further. Participatory inventories of benefits and risks, as well as a stringent adherence to ethics in practice will enable the responsible inclusion of marginalized populations in high quality qualitative research.

Reviewer 2

The lack of a keyword around the inclusion/perceptions of risk and benefit issues seems to be a gap – if I was looking for research like this (and I do) how would I find this article? We have added the following keywords to the submission:

Inclusion in research

Research risk perception 

The title is a problem for me – is there too much emphasis on the (researchers’) process of recruitment here? It doesn’t really represent the paper, as ironically one of the things I am asking for is more information about the recruitment process. I think ‘New considerations for inclusion in research’ is rather generic, and the paper isn’t about substance use in general.

Can the authors find a way to say more clearly that it is a study into e.g., something closer to indicating - What the views of people with perinatal opioid use disorders (OUD) who have experienced incarceration can tell us about the benefits and risks of participation in qual research interviews to inform ethical inclusion ... We have changed the title as follows: 

Ethical Inclusion: Risks and benefits of research from the perspective of perinatal people with opioid use disorders who have experienced incarceration 

I don’t need to say that it is well-written, but it is.

I like the subtle contribution to the non-identification of participants of ‘approximately half’ in each group.

I also like the seemingly effortless use of non-binary gendered language, which shows how easy it is do this.

I like the quality of the participant data, which tells us a lot about the research team.

And I like that the same code is identified as both a risk and a benefit (Helping Others. Making Change)

I very much like the exploration of what we might call the ‘advocacy/change/impact misconception’ (akin to the ‘therapeutic misconception’ in clinical research.) This is really important, particularly as it is something researchers’ can affect (take responsibility for) in the way we present studies. Gosh – thank you so much. We are thrilled that the study resonated with you. 

It took me a while to work out that this paper was reporting the whole study – and that it was not an adjunct to a study ‘about’ something else, that had led to gathering research participants’ views on that experience. This is great, and quite innovative I think – we certainly need more robust research (as this is) about this area. But this needs to be a bit clearer (including in the abstract and title) – this is important not just for clarity but because it would highlight that this is a more unusual paper, especially in this criminal justice setting. Thank you! In fact, in the Background of the ABSTRACT, we mention the aim of the study as such. However, there are additional research questions related to the project that we hope to explore in future papers. Therefore, we will not claim that this (while the first research objective of the study) was the sole purpose of our data collection efforts. 

There are various places where I thought, ‘But what did they tell the participants they were doing?; What questions did they ask in the interviews?’. Some of this is the inevitable ‘wanting to know more’ when you like something, but it is also about the presentation of the content so readers can orient ourselves to this study and its results. (See below). Yes – this is a good point. We added a section to lay out our interview questions more explicitly. MATERIALS AND METHODS

Interviews and Safety Monitoring

…

The main objective of the interview questions was to inform robust recruitment and community retention processes for people with OUD from a prison prenatal clinic. As such, the interview questions asked specifically how a participant heard about the study; why they chose to participate in the interview; what things were considered during that decision-making process; what, if anything would have made them want to participate more or less; and whether they would be willing to participate in a follow-up interview – if that were a possibility. While they were explicitly told that the current study did not include follow up interviews, they were asked how study staff might best contact them after transition back into the community, in the hypothetical case of a follow up interview.

I think the authors have two options – either to present this study more clearly as a specifically local/national example, as research in the US via an IRB, with various limitations and implications for improving the situation in the US, which could then be discussed in an international context, or 2. to present the study as situated in the US, but having implications globally – I would prefer the latter, and assume that was your intention, but it currently sits too far in the middle. Yes – We appreciate this point and this perspective. We do understand that literature is perhaps stronger when situated in a global context. 

However, there are two things that convince us, in this instance, it is the better choice for us to situate this study as more specifically a U.S. example. First, there are considerable differences distinguishing the U.S. carceral system – the largest in the world. It would be imprudent for us to extrapolate, given our limited understanding of prisons in other national contexts. Second, the data we collected is contextualized and understood only in its U.S. context. For that reason, we cannot generalize beyond. We have clarified at several points in the introduction and discussion that this has a U.S. focus and have also added this to the limitations. Introduction:

In the United States, existing research ethics guidelines as well as an emphasis on representation assist in structuring the inclusion of these populations; however, insights from the affected populations regarding the benefits and barriers to research participation are not adequately considered.

In the US, statute restricts research of people confined in prisons to studies designed to investigate phenomena or experiences affecting confined individuals as a distinct population or that has a reasonable probability of improving their wellbeing (Lapid, Ouellette, Drake, & Clarke, 2023).

Recent inquiries into the ethical considerations of research, particularly with marginalized populations in the US, have shown discrepancies between how participants view their participation in research studies and how they are typically presented in ethics review boards or study protocols (7).

We contribute to this important literature by relaying the perceived risks and benefits of research participation among a US population of individuals who have experienced the intersecting vulnerabilities of pregnancy, incarceration, and substance use disorder.

Discussion:

Thus, our findings should be interpreted in the context of the North Carolina institutions in which they were conducted – a perinatal treatment program and a state prison – even as they raise questions and considerations for a broader context.

The most prominent issue for me is the specificity/limitations of the paper regarding ethics governance in the US. This needs to be addressed for accessibility and usefulness for an international audience, but is straightforward to do.

For example the first sentence of the Introduction – does this whole sentence refer to the situation in the US, or is it only the final clause about incarcerated individuals in the US? I would like to see this setting/starting point more clearly framed from the very beginning. This is a good point and was not carefully done. We are opting to situate this analysis specifically in the US, and will make the suggested changes to clarify this choice. 

See above for the additional changes regarding US-specificity. INTRODUCTION

Research has proliferated over the last 25 years and increasingly includes women, pregnant individuals, individuals with substance use disorders, and individuals who are incarcerated in jails and prisons. In the United States, existing research ethics guidelines as well as an emphasis on representation assist in structuring the inclusion of these populations; however, insights from the affected populations regarding the benefits and barriers to research participation are not adequately considered.

The Introduction could use a little tweaking to clarify which parts/claims are about the US specifically, and where/at what point it broadens to the wider / international situation. It invokes a lot of pertinent international literature, although I note not much that is very recent and the debate is moving fast (choice of action on that left to the authors). It would be helpful to clarify the specific/generic nature of the argument here, and at what point this shifts from one to the other. Simple signposting would do the trick. We have tried to indicate throughout that we are focusing our exploration of the topic (and our particular study) in the US. We have cited a more recent international commentary and made a point that, while extrapolation to other contexts by international scholars may be of some use, it is beyond our scope in the current piece. 

There is a strong dependence on the Belmont Report in framing the paper, which does not have the same function beyond the US (although it is a standard authoritative reference of course). This is more of a comment, which the authors can consider as they wish. We do want to establish the setting and history of the research governance structure within which we operated, collected our findings, and are resultingly commenting on, hence the focus on the US history and structure. We do hope that this piece becomes a (albeit narrowly American) contribution to an important global conversation. Thus, our findings should be interpreted in the context of the North Carolina institutions in which they were conducted – a perinatal treatment program and a state prison – even as they raise questions and considerations for a broader context.

I do want to ask if there are any guidelines in the US regarding research in this setting – statutory/professional/disciplinary etc? Is the Belmont Report the only point of reference for the guidelines mentioned (the rest seems to be academic literature, or possibly guidelines nested in that academic literature)? This is a helpful question. We have added a citation to our introduction INTRODUCTION

In the US, statute restricts research of people confined in prisons to studies designed to investigate phenomena or experiences affecting confined individuals as a distinct population or that has a reasonable probability of improving their wellbeing (Lapid, Ouellette, Drake, & Clarke, 2023).

The current recommendations/comments regarding governance have unacknowledged limitations, as there are different practices and expectations in other countries. Also different terminology regarding e.g., Research Ethics Committees as opposed to IRBs. Simply using a combination term (RECs/IRBs) would open this up. Yes – we have made the recommended expansion here and also revised the discussion to make this more inclusive of other research ethics structures. INTRODUCTION

The Belmont Report has guided determinations of ethical research practices through institutional review boards and research ethics committees with its basic moral principles of respect for persons, autonomy, beneficence, and justice (1, 2).

DISCUSSION

Research into the experience of those being interviewed differed in the assessment of risk and benefit from normative ethics review protocol language, underscoring the importance of ascertaining subjective perspectives of research participants during the study protocol (21).

Some of the specific ethics/governance claims about IRB requirements would not apply in other countries, e.g., (381-83). I am not aware of any research situation internationally with a blanket ban on social media recruitment, in fact quite the opposite is true, especially since 2020. So this needs contextualising (was this perhaps study-specific?). 

Yes – we agree this differs in other settings and has particularly changed since 2020. We have softened our language, while still finding the merit in mentioning the (potential) barrier, and clarified why the requirements that often remain (i.e., to obtain consent for unencrypted communication via phone or in person) may still cause problems. DISCUSSION

For example, prohibitions on the use of social media for study communication without prior written or verbal consent via another communication medium may directly limit participation for those without stable addresses or phone numbers.

And contra the situation described in the paper, in UK ethics review (and in many other countries and regarding international research codes), researchers are expected to consider the benefits and risks that potential research participants from all groups should be made aware of in order to give informed consent (90-93). This has led to the standard inclusion of ‘Patient and Public Involvement’, community consultation etc (or equivalent)s in research design, which is now a recommendation/requirement of many funders across the world. It is nowhere near perfect or adequate, but this situation doesn’t chime with the statements in the paper that there is no expectation on researchers to do this. Can the authors make it clear what they mean here? Ah, thank you for telling us about this. Yes, despite some movement in the US, particularly in the field of community-based participatory research and patient-centered research design, the reductionist approach of most ethics review committee processes in the US led us to use this framing. We have tried to tighten our language here to make this more clear. INTRODUCTION

Institutionalized ethics procedures run the risk of becoming perfunctory, without considering, from the perspectives of those who have been historically underrepresented or excluded, the unique benefits and risks of which potential research participants should be made aware in order to give informed consent (Hammett, Jackson, & Bramley, 2022).

And would any references to the international literature (codes/guidelines) help? If you feel not, then don’t do it.

Does the situation reported relate solely to the individual IRB approach for this study? Or is this more general in USA? This is of real interest to a wider readership, so even a couple of sentences would help to situate this. Thank you – I think, in this case, it would be more consistent with our piece to situate our introduction more firmly in its US context. We have added language specific to that context in this place that you suggest. 

See above for additional modifications. INTRODUCTION

In the US, statute restricts research of people confined in prisons to studies designed to investigate phenomena or experiences affecting confined individuals as a distinct population or that has a reasonable probability of improving their wellbeing (Lapid, Ouellette, Drake, & Clarke, 2023). However, a tension exists between protectionism, or restricting the study of populations that are deemed vulnerable to coercion, and fair inclusion in ethical research (Friesen et al., 2023).

What was the informed consent process, and what were participants told?

Informed consent processes are discussed several times, both generically, and specifically for this study. But I would like to be clearer about what material this contained and what the participants’ journeys were. We have added a section describing informed consent for this study. Following screening and enrollment, participants were offered time prior to consenting to participate in the study. The informed consent process was conducted via Webex phone or video by the study coordinator (JJ). For participants who were incarcerated at the time of the interview, an additional non-clinician member of the research team observed their Webex use inside the facility in a room where they could not be observed or overhead by carceral facility staff. Participants who were not incarcerated were instructed to call or connect via video from a private location. The study coordinator reviewed the details of the study and the consent form and answered questions. All potential participants were informed that their participation would not affect prenatal care during or after incarceration or any aspect of parole or incarceration. For participants at NCCIW, the additional non-clinician staff member witnessed the participant’s signature on the consent form. Participants not at NCCIW provided verbal consent as all study procedures were conducted via video and phone. Of note, in order to avoid a real or perceived conflict of interest, the senior author was not involved in the consent process as she is also a prenatal care provider at the prison.

The incarcerated participants received a flyer, but what did it say? How did they opt in following that? There is a subsequent process, but we don’t know what it is. In a context of incarceration discussing the autonomy of participants, this is crucial to understand. This connects to the data about participants’ decision-making processes, which would be stronger if we knew the context of what that process included for them. We have revised the “Recruitment and Enrollment” section to include additional details of how they opted in. We recruited participants for the study at both NCCIW and the treatment program between September 2021 and April 2022. Recruitment at NCCIW consisted of providing clinic nurses with postcard-sized flyers to share study information. Potential participants who were interested in learning more about the study were directed to a private space in the prenatal clinic which they could ask study staff questions about the study; this interaction occurred either just before or just after a prenatal visit such that clinic and facility staff would not be able to distinguish between receiving usual clinical services or learning more about the study. At the treatment program, we presented the study very briefly at several group sessions. Information at these sessions was similar to the flyers. Potential participants were directed to contact the study team directly to learn more about the study. We did not enumerate the number of people outreached to who declined to participate. Currently incarcerated participants did not receive any financial incentive for participating – per prison policy – and non-incarcerated participants were compensated with $50 for their interview.

What does the consent form discussed in this study contain? In UK (again as an example but this is common practice elsewhere too) the standard process is that a consent form is a separate document which cross-refers to the Participant Information Sheet (usually constructed against a standard template), which includes all relevant details. So if someone only saw the consent form, they would not have much information. I think some simple more explanatory information here would be helpful for a wide international readership. We have added additional details about the consent form in the new “Consent” section. The study coordinator reviewed the consent form, which described details of the study as well as potential risks and benefits of participation, and answered questions.

The treatment programme participants had a different route into the study. Do we need to know anything more about this differentiation as it seems important? What did the presentations entail? Might it have impacted their understanding of the research? It sounds as though it was more interactive and discursive. The presentations were very similar to the flyers in terms of content. Our objective was to distract minimally from the treatment group and to provide additional information on a one-on-one basis for interested participants. We have revised the recruitment paragraph to reflect this. We recruited participants for the study at both NCCIW and the treatment program between September 2021 and April 2022. Recruitment at NCCIW consisted of providing clinic nurses with postcard-sized flyers to share study information. Potential participants who were interested in learning more about the study were directed to a private space in the prenatal clinic which they could ask study staff questions about the study; this interaction occurred either just before or just after a prenatal visit such that clinic and facility staff would not be able to distinguish between receiving usual clinical services or learning more about the study. At the treatment program, we presented the study very briefly at several group sessions. Information at these sessions was similar to the flyers. Potential participants were directed to contact the study team directly to learn more about the study. We did not enumerate the number of people outreached to who declined to participate. Currently incarcerated participants did not receive any financial incentive for participating – per prison policy – and non-incarcerated participants were compensated with $50 for their interview.

Could any of this recruitment material go in as supplementary material (IRB approval allowing but there should be no reason why this couldn’t be shared)? That would make it easy to just add small points where relevant and cross-refer to the supplementary material. We have included our informed consent document as a supplemental appendix (Appendix A). 

It would be good to know what the questions for the interview were – e.g., were the participants asked about other experiences of research (as suggested by some of the data)? A couple of sentences would be enough, with or without the schedule as supplementary material. We have revised our description of the interview guide. We have included it as an appendix; although it includes a list of questions, the specific wording likely varied slightly with each interview based on the interaction between the interviewer and interviewee. 

 MATERIALS AND METHODS

Interviews and Safety Monitoring

The interview guide was designed from a trauma-informed perspective, rooted in a Reproductive Justice theoretical framework (14, 15), and included important domains from the literature and the research questions. The main objective of the interview questions was to inform robust recruitment and community retention processes for people with OUD from a prison prenatal clinic. As such, the interview questions asked specifically how a participant heard about the study; why they chose to participate in the interview; what things were considered during that decision-making process; what, if anything would have made them want to participate more or less; and whether they would be willing to participate in a follow-up interview – if that were a possibility. While they were explicitly told that the current study did not include follow up interviews, they were asked how study staff might best contact them after transition back into the community, in the hypothetical case of a follow up interview.

It would be good to know what reasons participants were given to take part in the study/what the agenda was – and to know how that intersected with their decision-making and the data that emerged. For example on p11 (264-5) there is a reference to the research goal, but we don’t know what that was. Or how it was communicated to potential participants. The full description of what was given to participants is in the consent form (Appendix A) and we have added a short description to the “Consent” section of the methods. 

We have also revised the results to include additional context. CONSENT

Participants were informed that the goal of the research was to learn about the challenges facing women who use drugs and are pregnant or just had a baby while they are in prison, and specifically that the study aimed to understand what makes women decide to be in a research study or not to be in a research study, and what might make it easier for them to stay in a study when they go home.

RESULTS

Risks

Misunderstanding the scope and pace of the research

A key risk was related to how some participants described their understanding of the purpose of the study in the context of their motivation to participate. Although the consent process included a description of the study that emphasized learning about challenges facing participants and understanding their decisions to participate or not in research generally, some of their responses implied confusion between advocacy and research in terms of the scope and pace of change that could come from participation. Multiple interviewees talked about participating out of a desire to make change, which was described above as a benefit of participation. However, some participants spoke of their participation in the study as an opportunity to advocate or make change more directly and in a shorter timeframe. 

343-346 – This is just a comment from practice: I am curious about why there would not be an option for the participants to contact the researcher regarding future studies. It is standard practice in most settings for participants to be given the lead researcher’s (at least) contact details before they even opt-in, and although in practice people rarely use these, they are generally urged to keep the information for their records. So referring to this would be a straightforward way to keep lines of communication open (although the only time I tried it, no-one contacted us even though they said they would …). It may of course not be possible/feasible for incarcerated people to receive/keep this information, but those in e.g. a treatment programme might appreciate it? It would not resolve all the issues of course, but my point is that it is an existing mechanism which could help. No action is required here. Thank you for this point! We definitely provided contact information to participants. We agree that it is unclear whether it was feasible for incarcerated participants to keep this information or act upon it in future. 

In my experience, especially in social-science/health type studies, much participant information refers to the benefit of sharing, helping others, improving services etc. Often in ethics review we have to roll this back a bit, to not over-promise to participants the realistic impact of their contributions. Again, is this context-specific? If there are such limitations where the authors are working this is of huge interest to others. Yes, we intend here to say that participants’ reflected motivations for participation were outside of what was is generally considered typical or appropriate by our ethics review committee. We similarly would not want to overpromise regarding potential benefits to individuals and society. We do not want to be too heavy handed with this claim, but are glad it may have resonance in other contexts. 

I’m not sure what ‘ascertaining subjective perspectives of research participants during the study protocol’ means (359-60). Is something missing here? Ah – obtuse syntax. We have reworded. DISCUSSION

Research into the experience of those being interviewed differed in the assessment of risk and benefit from normative ethics review protocol language, underscoring the importance of incorporating the perspectives of research participants during the study protocol (21).

Knowing what participants were told (as above) would help consideration of their understanding of the boundary between research/advocacy as it would provide some context when we get to this central point. Yes, hopefully by including the text and appendix above this point is more clear. 

I’m confused about why there is a suggestion to extend ‘capacity’ to situational factors. Capacity is generally understood as literal clinical type capacity to understand, and there is quite rightly much discussion about e.g., not excluding people with dementia, or those with learning disabilities from research on this basis. I doubt it is intended, but currently this point sounds like it plays into the hands of paternal exclusionists, or ‘protectionism’ (385), which the authors say they want to resist, and I am not sure what purpose it serves here. Given the international debate about challenging concepts of vulnerability in research mentioned (388-90 which again seems maybe light on the standard/recent literature, but again is the authors’ decision to act or not) this seems out of step – do you need this here at all? Or if you do want to include it, can you clarify what you mean? Ah – this is a very good catch – thank you. We mistakenly interpreted capacity in a more colloquial sense (of being available to do something, not in a clinical sense). We have corrected this. 

We have also added an updated citation for the point about vulnerability vs inclusion. 

 DISCUSSION

The process of informed consent relies on voluntariness, disclosure, understanding, and capacity (23). In addition to these four principles, we propose the consideration of availability to participate. Our findings highlight the ways that freedom to participate in research is compromised in the lives of marginalized populations, and challenges research committees to encourage equitable ways to reduce logistical barriers to participation for those unduly burdened.

As a tiny point (361) It would be helpful just to clarify if there was actually a risk of breach of confidentiality (which I doubt) or was it a perceived (highly justifiable) concern for participants? This is a good point; thank you. DISCUSSION

Risks identified in this study included fear of a breach of confidentiality, and participants shared complicated decision-making about how much this risk weighed in their decisions.

---

## [Decision Letter · Decision Letter 1]

10 Oct 2023

PONE-D-23-02896R1Ethical Inclusion: Risks and benefits of research from the perspective of perinatal people with opioid use disorders who have experienced incarcerationPLOS ONE

Dear Dr. Reddy,

Thank you for submitting your manuscript to PLOS ONE. After careful consideration, we feel that it has merit but does not fully meet PLOS ONE’s publication criteria as it currently stands. Therefore, we invite you to submit a revised version of the manuscript that addresses the points raised during the review process.

We look forward to receiving your revised manuscript.

Kind regards,

Alan Farrier, Ph.D

Academic Editor

PLOS ONE

Journal Requirements:

Additional Editor Comments:

Well done on the revisions to this paper! It has just come back from the second review with a few minor comments to address from one of the reviewers.

Reviewers' comments:

Reviewer's Responses to Questions

**Comments to the Author**

1. If the authors have adequately addressed your comments raised in a previous round of review and you feel that this manuscript is now acceptable for publication, you may indicate that here to bypass the “Comments to the Author” section, enter your conflict of interest statement in the “Confidential to Editor” section, and submit your "Accept" recommendation.

Reviewer #1: (No Response)

Reviewer #2: All comments have been addressed

2. Is the manuscript technically sound, and do the data support the conclusions?

Reviewer #1: Yes

Reviewer #2: Yes

3. Has the statistical analysis been performed appropriately and rigorously? 

Reviewer #1: N/A

Reviewer #2: N/A

4. Have the authors made all data underlying the findings in their manuscript fully available?

Reviewer #1: Yes

Reviewer #2: No

5. Is the manuscript presented in an intelligible fashion and written in standard English?

Reviewer #1: Yes

Reviewer #2: Yes

6. Review Comments to the Author

Reviewer #1: The authors have done a great job incorporating feedback from the first review. The manuscript is significantly improved. I have a few minor comments:

-The first paragraph of the manuscript reads as overly broad and I'm not sure that it is necessary to start the paper. If the authors choose to retain this paragraph, then there should probably be citations included for the first and second sentences in this paragraph -- if this is "common knowledge" then I'm not quite sure it's necessary to include any of this text to start the paper.

-Please spell out United States and give the abbreviation and then use the abbreviation from that point forward consistently.

-Check grammar/sentence structure of the first sentence under eligibility and consent (p. 5, line 131)

-Two subheadings now refer to consent (pp. 5 & 6). I would remove 'consent' from p. 5 (line 130).

-The sentence re: payment feels out of place (p. 6, line 158-160) because participants would be compensated after they consented. I would move this sentence to the last sentence before analysis (p. 8).

-On Page 8, lines 190-191 -- I think you need to make it clear that the main objective of the interview questions that *were used in this study* were to inform robust recruitment and community retention processes. It's clear from the Appendix B that there were other questions included that were there for other purposes (and not presented in this paper). I would just try to make that clear that for *this* paper/study, you pulled out specific questions (Sections 2 & 4, specifically) from the broader interview protocol.

-There are weird lines that appear throughout the paper and I'm not sure if they're intended to be there (e.g., p. 9, 14, 15).

Reviewer #2: Just to say I have nothing else to add to a review. I really like this paper. I am glad to have been able to support it, and look forward to seeing it in print (and being able to cite it).

I would also like to thank the authors for the positive responses to the review and engagement with it. The changes have made it better.

Do you realise you have written an empirical ethics paper? That may be worth you exploring ...

7. PLOS authors have the option to publish the peer review history of their article (what does this mean?). If published, this will include your full peer review and any attached files.

Reviewer #1: No

Reviewer #2: **Yes: **Julie Cook

---

## [Author Response · Author response to Decision Letter 1]

1 Nov 2023

Reviewer #1:

1. The first paragraph of the manuscript reads as overly broad and I'm not sure that it is necessary to start the paper. If the authors choose to retain this paragraph, then there should probably be citations included for the first and second sentences in this paragraph -- if this is "common knowledge" then I'm not quite sure it's necessary to include any of this text to start the paper. 

***Ah, you are correct. The information in this paragraph is stated, more specifically and with citations, later in the Introduction. The intention was to create a summary “here’s what this paper is”, but you are correct that it is overly broad and redundant with the abstract. We have removed. (Lines 65-71 removed.)

2. Please spell out United States and give the abbreviation and then use the abbreviation from that point forward consistently. 

***Thank you! Yes, we have made these corrections throughout. (Fixed on Line 44; 96; 105; 126. We also corrected our abbreviation of North Carolina on lines 44 and 142.)

3. Check grammar/sentence structure of the first sentence under eligibility and consent (p. 5, line 131) 

***Thank you; yes, this was oddly phrased. We have removed ‘currently’ as ‘past year’ encapsulates that timeframe. Hopefully this helps. (“We recruited people who were at least 18 years old, met the clinical criteria for opioid use disorder (OUD), and had experienced perinatal incarceration in the past year.”)

4. Two subheadings now refer to consent (pp. 5 & 6). I would remove 'consent' from p. 5 (line 130).

***Good catch! We have removed. Thank you. (“Consent” removed as section heading (page 5; line 130))

5. The sentence re: payment feels out of place (p. 6, line 158-160) because participants would be compensated after they consented. I would move this sentence to the last sentence before analysis (p. 8).

***Great suggestion! We have moved that sentence to the end of the “Interviews and Safety Monitoring” section(Sentence moved to line 199-201.)

6. On Page 8, lines 190-191 -- I think you need to make it clear that the main objective of the interview questions that *were used in this study* were to inform robust recruitment and community retention processes. It's clear from the Appendix B that there were other questions included that were there for other purposes (and not presented in this paper). I would just try to make that clear that for *this* paper/study, you pulled out specific questions (Sections 2 & 4, specifically) from the broader interview protocol.

***This is a good distinction. Our Primary Study Aim was recruitment and retention, however, we did have subaims, which are reflected in the other research questions. Your suggestion is the most integrous to the interview protocol, therefore we have added the language you recommend. (Line 192-194: “The main objective of the interview questions that inform this analysis was to inform robust recruitment and community retention processes for people with OUD from a prison prenatal clinic.”)

7. There are weird lines that appear throughout the paper and I'm not sure if they're intended to be there (e.g., p. 9, 14, 15).

***Oh, those lines! We tried everything to remove them – stubborn page breaks. Anyway, in the Manuscript attachment, I have copied our text over to a clean document and they are (finally) gone. Thank you for noticing that! (They still appear in the Track Changes document, but have been removed from the Manuscript attachment.)

Reviewer #2: Just to say I have nothing else to add to a review. I really like this paper. I am glad to have been able to support it, and look forward to seeing it in print (and being able to cite it). I would also like to thank the authors for the positive responses to the review and engagement with it. The changes have made it better. Do you realise you have written an empirical ethics paper? That may be worth you exploring ... 

***Thank you so much. Your careful read and thoughtful feedback improved our work; we greatly appreciate that. We do hope it will be a useful resource for those doing this important work. Thank you.

---

## [Editor Report · Decision Letter 2]

6 Nov 2023

Ethical Inclusion: Risks and benefits of research from the perspective of perinatal people with opioid use disorders who have experienced incarceration

PONE-D-23-02896R2

Dear Dr. Juila Reddy,

We’re pleased to inform you that your manuscript has been judged scientifically suitable for publication and will be formally accepted for publication once it meets all outstanding technical requirements.

Kind regards,

Alan Farrier, Ph.D

Academic Editor

PLOS ONE
---

## [Editor Report · Acceptance letter]

8 Nov 2023

PONE-D-23-02896R2 

Ethical Inclusion: Risks and benefits of research from the perspective of perinatal people with opioid use disorders who have experienced incarceration 

Dear Dr. Reddy:

I'm pleased to inform you that your manuscript has been deemed suitable for publication in PLOS ONE. Congratulations! Your manuscript is now with our production department. 

Kind regards, 

on behalf of

Dr. Alan Farrier 

Academic Editor

PLOS ONE